# Advancing Archaeobotanical Methods: Morphometry, Bayesian Analysis and AMS Dating of Rose Prickles from Monteagudo Almunia, Spain (12th Century–Present)

**DOI:** 10.3390/plants14243709

**Published:** 2025-12-05

**Authors:** Diego Rivera, Julio Navarro, Inmaculada Camarero, Javier Valera, Diego-José Rivera-Obón, Concepción Obón

**Affiliations:** 1Departamento de Biología Vegetal, Facultad de Biología, Universidad de Murcia, Campus de Espinardo, 30100 Murcia, Spain; javier.valera.martinez@gmail.com; 2Laboratory of Archaeology and Architecture of the City, Escuela de Estudios Árabes (CSIC), Cuesta del Chapiz, 22, 18010 Granada, Spain; julionavarro@eea.csic.es; 3Contrato Investigador con Cargo a Proyecto, Departamento de Historia Medieval y Ciencias y Técnicas Historiográficas, Facultad de Filosofía y Letras, Campus Universitario de Cartuja, 18071 Granada, Spain; inmacamcastellano@gmail.com; 4RITM (Research Center in Economics & Management), Université Paris-Saclay, 54, Boulevard Desgranges, 92330 Sceaux, France; diego-jose.rivera-obon@universite-paris-saclay.fr; 5Instituto de Investigación e Innovación Agroalimentario y Agroambiental (CIAGRO), Escuela Politécnica Superior de Orihuela, Universidad Miguel Hernández, Ctra. Beniel, Km 3.2, 03312 Orihuela, Alicante, Spain

**Keywords:** garden archaeology, medieval archaeology, post-medieval archaeology, Muslim Almunia, *Rosa* L., rose prickles, Western Islamic medieval rose heritage

## Abstract

Background: While archaeological evidence is crucial for understanding the origins of ancient rose varieties in Western Europe, the botanical composition of medieval Islamic gardens remains largely unknown. This study focuses on the rose cultivation at the 12th-century Almunia del Castillejo de Monteagudo in Murcia, Spain, a key Islamic site in al-Andalus. Methods: Morphometric analysis and Bayesian hypothesis testing were applied to characterize rose prickle remains recovered from the site. The prickles were found in stratigraphic contexts above the original garden and yielded post-medieval radiocarbon dates (18th–19th centuries AD). The morphological parameters of the archaeological specimens were statistically compared against reference collections of known rose species to determine their probable botanical origins. Results: The analysis identified two distinct prickle morphotypes. Statistical comparisons indicate these correspond to the white musk rose (*Rosa moschata* Herrm.) and to yellow roses from the *Rosa foetida* Herrm. complex (including *R. lutea* Mill. var. *persiana* Lem.). Both species are historic introductions from West Asia. The morphometric parameters demonstrated significant diagnostic value for the species-level identification of archaeological rose remains. Conclusion: Despite the post-medieval date of the prickles, the presence of *R. moschata* and *R. foetida* suggests the continuity of cultivation for rose species originally already known during the medieval Islamic period. This provides direct archaeological evidence for the role of al-Andalus gardens in the dissemination of West Asian rose diversity, highlighting the lasting impact of medieval Islamic horticulture on the Iberian Peninsula.

## 1. Introduction

Archaeobotanical studies face two fundamental challenges: botanical identification of remains and their chronological placement or dating. Adequate resolution of both issues enables meaningful interpretations of potential uses, cultural significance, and landscape evolution through time to the present day. This requires supplementary methodological approaches, including spatial distribution analysis of remains, taphonomic assessment, and examination of the complete stratigraphic sequence, rather than focusing solely on periods of primary interest. Traditional reliance on expert visual identification, preferential treatment of charred over desiccated material (considered suspect), and artifact-based dating has long demonstrated inherent limitations. Radiocarbon dating of organic remains weighing only a few milligrams, combined with morphometric techniques employing machine learning or Bayesian inference, provides pathways toward more accurate sequence reconstruction. We apply these approaches here to rose prickles recovered from Monteagudo.

### 1.1. Roses in the Archaeobotanical Register

Archaeobotanical evidence for roses is based on pollen, seeds, fruit fragments, wood and prickles. Pollen [1,2,3] is referred to as *Rosa* sp.; however, when it is found along large chronological sequences, it is difficult to discern which is original and which is a result of contamination. In the case of wood fragments [1,2,3], these can be individually radiocarbon dated, and their taxonomic identification relies not only on morphometrics but also on fine analyses of cell patterns using Scanning Electron Microscopy.

### 1.2. Background on the Rose Diversity

#### 1.2.1. Rose Diversity and the Role of Hybridization

The morphological diversity of rose flowers (UPOV (2006), CPVO (2009), and GOV.UK (2011) [4,5,6]), together with their perfume and notably, their colors, as well as the repeated blooming within the same vegetative season, which are the main characteristics of cultivated roses, have attracted the attention of the various cultures that have succeeded each other in the region since their introduction in the Mediterranean. In this area, some cultivated rose species and hybrids are native autochthonous, but others made their debut in the early first millennium BCE, initially as exotic commodities reserved for the elite, featuring prominently in religious rites. The genetic diversity of cultivated roses is manifested not only in their pedigree but also in the different ploidy levels of the hybrid strains, ranging from diploid to hexaploid [7,8,9].

To understand the potential emergence of ornamental hybrids of fragrant roses, such as *R*. × *bifera* (Poir.) Pers., *R.* × *damascena* Herrm., *R.* × *centifolia* L., and *R.* × *alba* L., in Mediterranean gardens, it is essential to examine their ancestral species. Among the seven basic ancestral species—*Rosa canina* L.*, R. pulverulenta* M.Bieb. (syn. *R. sícula* Tratt.), *R. corymbifera* Borkh., *R. gallica* L., R., *R. webbiana* Wall. ex Royle (syn. *R. fedtchesnkoana* Regel)*, R. phoenicia* Boiss., and *R. moschata* Herrm.—only the first four have been present in the Mediterranean region for millennia. The last three species extend over different regions of West Asia. Consequently, hybrids like *R.* × *bifera*, *R.* × *damascena*, and *R.* × *centifolia* originated in the border areas between West Asia and Mediterranean Europe. Provided the theory that *R.* × *alba* is a hybrid of *Rosa corymbifera* and *R. gallica* is right, it could be the only fragrant hybrid rose that originated entirely within the Mediterranean region [10,11,12,13].

Regarding the diversity of the tetraploid *Rosa* × *damascena*, Kiani et al. [14] found that the germplasm of this species in Iran is quite varied. A sample from Bulgaria falls within the main Iranian group, suggesting that *R.* × *damascena* in Bulgaria and Turkey was obtained from rose production areas in the provinces of Fars or Isfahan in Iran. The high diversity of this species in Iran indicates that the region could be a major center of its diversity. This suggests that ancestral *R.* × *damascena* was introduced to the Mediterranean from Persia during the campaigns of Alexander the Great in the fourth century BCE.

*Rosa foetida* Herrm. is tetraploid, with five to ten yellow petals, and was cultivated in medieval Western gardens. Although Columella’s mention of yellow roses could have referred to it, it was often grown alongside or followed by the double-flowered *R. lutea* Mill. var. *persiana* Lem. *Rosa foetida* has played a crucial role in the breeding of hybrid roses; the deep yellow color from *R. foetida* ‘Persian Yellow’ was first transferred to modern roses by Pernet-Ducher in 1898. The first yellow cultivar, ‘Soleil d’Or,’ introduced in 1900, became the common ancestor of all subsequent yellow cultivars. The bright yellow and orange hues in Hybrid Tea roses originate from this single breeding event [15].

Studies of microsatellite markers (SSR) by Samiei et al. [15] show the close relationship between the red-yellow rose or Austrian briar (*Rosa bicolor* Jacq.) and *R. foetida*; both are tetraploids. They also reveal that *R. lutea* var. *persiana* forms a separate group and that yellow roses cultivated in Western Europe, previously identified as *R. foetida*, are hybrids with other *Rosa* species.

#### 1.2.2. Roses in Classical Antiquity and Middle Ages of Europe

The history of cultivated roses in Mediterranean Europe, prior to the present period, can be delineated into four principal chronological phases. The first phase encompasses Classical Antiquity, spanning from the 8th century BCE to the 5th century CE. This is followed by a second phase that extends from Late Antiquity to the High Middle Ages, covering the 6th to the 16th century CE.

Classical Antiquity roses were based on a limited repertory of local wild rose species, such as *Rosa gallica* and *R. pulverulenta,* but also relied upon the imported roses from Persia and neighboring areas of West Asia, such as *Rosa* × *damascena*, *R.* × *centifolia* or *R.* × *bifera* [16,17].

Seemingly, Medieval Rose diversity originated from this classical heritage but also from novel introductions from Persia, the Caucasus and Anatolia, and local hybridization events. Western Muslim gardens seem to have played a key role. Ibn Luyūn (1282–1349 CE) describes the optimal structure of a *bustān* for enjoyment and utility. He suggests: “After the *zafariche* [small artificial pond], plant evergreen plants. … In the center, build a pavilion (*qubba*) with views in all directions, surrounded by rose bushes and myrtles to beautify the space. The entire orchard (*bustān*) should be enclosed by a high wall for protection and privacy” [18]. In marginal Note 6 [v. 14], Ibn Luyūn lists plant species ideal for the *bustān*, including ornamental shrubs, six types of citrus trees, aromatic plants, and flowers near the pavilion or leisure area and rose bushes (*ward*) (*Rosa* sp.pl., notably *Rosa* × *damascena* Herrm.), and *Rosa moschata* bushes (*ward al-zīna*), among other ornamentals [18].

Roses were cultivated in the Nasrid gardens of Granada. In the Court of the Water Channel at the Generalife (Granada, Spain), palynological analysis of medieval layers identified 47 diverse types of pollen. These included myrtle, cypress, three types of citrus trees (bitter orange, lemon, and citron), roses, laurels, and jasmine [18,19].

Andalusian authors distinguished numerous species and varieties of roses based on the color of the flowers, the number of petals, and their original habitat (Table 1).

The third phase extends from the Late Middle Ages through the Baroque Period into the early Enlightenment, spanning from the 14th century CE to the mid-18th century CE. This period is particularly notable for significant horticultural developments, including the Dutch cultivation of up to 200 improved strains of *Rosa* × *centifolia* and the emergence of Moss roses. The latter originated prior to the mid-sixteenth century as a sport of *R.* × *centifolia.* Especially the yellow varieties, which were well-known since the Renaissance, but only occasionally mentioned in Roman and Western Muslim texts [16,17]. The case of *Rosa moschata* is paradigmatic since it was widely grown in Spanish gardens of Sevilla and Granada in the 16th century CE. It was cultivated for its essence in North Africa for centuries and would be the Cyrenaic roses mentioned by the Romans. Furthermore, it participates in the hybrid origins of *R*. × *bifera* [26].

#### 1.2.3. The Chinese Roses Revolution and Modern Roses

The Chinese roses revolution commenced with the cultivation of *Rosa chinensis* Jacq. “Old Blush” in England in 1781. This introduction catalyzed a series of subsequent hybridization events between 1790 and 1850 that produced several important rose classes: Portland roses (*R.* × *damascena* × *R. gallica* × *R.* × *chinensis* Jacq.), Bourbon roses (*R.* × *bifera* × *R.* × *chinensis* “Old Blush”), and Noisettes (*R. moschata* × *R.* × *chinensis* “Old Blush”). Hybrid perpetuals also emerged during this period as descendants of Bourbons through further hybridization [16,17].

The introduction of Tea roses from China occurred concurrently, with notable examples including *Rosa indica* L. var. *odoratissima* Lindl. A significant milestone was achieved with Rosa ‘La France’, a pink rose cultivar discovered in France in 1867 by the rosarian Jean-Baptiste André Guillot.

This period also witnessed the expanded cultivation of several other rose types: Evergreen roses derived from *Rosa sempervirens*, Scotch roses (*Rosa spinosissima* L.), and Sweet briars (*R. rubiginosa* L.) [16,17].

During the 1900s, additional rose types appeared, including Ayrshire roses—climbing roses derived from *Rosa arvensis* Huds.—and Wichuraiana hybrids. The development of cluster-flowered roses followed, encompassing Polyantha and Floribunda types, alongside numerous other modern rose categories such as Miniatures, Ground cover roses, Modern shrub roses, Patio roses, Ramblers, and English roses [16,17].

#### 1.2.4. Evidence for Modern Wild and Cultivated Rose Diversity in the Context of Monteagudo (Murcia) and Neighboring Areas

Charco et al. [27] documented seven autochthonous wild *Rosa* species within the Murcia region in 2015: *Rosa spinosissima* L., *R. sicula* Tratt., *R. micrantha* Sm., *R. canina* L., *R. corymbifera* Borkh., *R. deseglisei* Boreau, and *R. pouzinii* Tratt. These taxa, along with the taxonomically uncertain *R.* × *nitidula* Besser, had been previously cataloged by Sánchez et al. [28] from various mountainous areas throughout the region. Significantly, all species were found at elevations above 500 m, with none recorded at the lower altitudes characteristic of the “Vegas del Segura,” where Monteagudo is situated.

Additional historical records confirm this altitudinal distribution pattern. Alcaraz [29], in his comprehensive study of northeastern Murcia’s flora and vegetation, documented *Rosa canina* and *R. pouzinii* from deep valleys north of Sierra de la Pila, at 35 km from Monteagudo and altitude over 300 m. Similarly, Fernando Esteve [30] reported *R. canina* from Sierra Espuña at elevations of 1300 m.

The most intriguing historical records originate from nearby Orihuela. Willkomm [31] reported *Rosa* × *almeriensis* Rouy, citing: “*In regno Valent[ino], (pr. Orihuela, ROUY) et Granat[ense], orient[ale], (pr. Velez-Rubio in Sierra de Maimon, ROUY)*.” However, detailed examination of the original work in both available versions reveals that *R.* × *almeriensis* was not actually documented from Orihuela. The sole locality mentioned for this species was: “*Le long du ruisseau (barranco del Caballon [Cerro del Maimon Grande, Velez Rubio]) qui serpente sur les flanks du cerro croissent*: *Ononis procurrens* Wallr., *Rosa Almeriensis* Rouy,…” [32] (p. 22), [33] (p. 245).

More reliable records for the Orihuela vicinity exist for *Rosa agrestis* Savi. Serra [34] references a herbarium sheet by De la Torre and Baeza documenting this species from a north-facing ravine in Sierra de Callosa at 150 m elevation in 1993, approximately 30 km north of Monteagudo. More recently, Escudero et al. [35] discovered *R. agrestis* in 2018 near irrigation channels at 21 m elevation south of Orihuela along the road to Hurchillo, approximately 20 km from Monteagudo. This species represents the most probable wild rose to have been naturally present in the Huerta de Murcia, though it was frequently confused by taxonomic authorities with *R. sempervirens* L.

Historical evidence indicates that ornamental rose cultivars were cultivated in post-medieval elite home gardens throughout Murcia and Orihuela, as well as in public gardens. These included *Rosa* × *damascena* Herrm., *R.* × *centifolia* L., *R. foetida* Herrm., *R. chinensis* Jacq., and the distinctive *R. chinensis* f. *viridiflora* (Lavallée) C.K.Schneid., among others [36,37].

### 1.3. Rose Prickles Diversity

Prickles, epidermal outgrowths prevalent in many *Rosa* species, function as defensive structures against herbivory and environmental stressors. However, in the case of hooked prickles, they act as fasteners to help the climbing stems of some rose species attach themselves to the branches of the trees that support them. Their morphological and structural diversity offers significant taxonomic and evolutionary insights.

Most rose bushes have “prickles” on their stems. These are rigid, sharp trichomes originating from an epidermal swelling and lacking conductive vessels. In contrast with true thorns, prickles can be detached without tearing the stem’s rigid tissues. Unlike thorns, which are modified stems, or trichomes, which are single-celled projections or formed by a very small number of cells, prickles originate from multiple ground meristem cells beneath the protoderm. Prickles in *Rosa* are classified into two types: glandular prickles (GPs), which feature a secretory glandular head at the tip and are involved in chemical defense, and non-glandular prickles (NGPs), which lack glands and primarily serve as mechanical deterrents and climbing aids. Further subcategories are based on branching, hairiness, and lignification patterns [38,39,40,41].

NGPs undergo lignification, hardening over time, while GPs, “bristles,” exhibit glandular differentiation at the tip [40]. The NGPs vary in size from large, triangular ones with a base over 5 mm wide to “needles” 1 mm long and 0.2 mm wide.

The morphology of prickles, including shape, density, and glandularity, aids in species identification, distinguishing between wild and cultivated roses such as *R. multiflora* Thunb. and *R. lucieae* Franch. & Rochebr. ex Crép. (syn. *R. wichuraiana* Crép. ex Déségl.) ‘Basye’s Thornless.’ The LOG gene family, which regulates prickle formation across various species like roses and eggplants, suggests convergent evolution [42]. Prickle traits also serve as phylogenetic markers; for instance, glandular prickles are rare in modern hybrids but common in wild species like *R. indica* (syn. *R. cymosa* var. *puberula* T.T.Yu & T.C.Ku) [39,40,41,42,43,44]. In terms of curvature, there are straight prickles and curved or “hooked” ones, with some being slightly curved and narrow. The predominant types can vary within the same plant depending on the stem type and may also appear in combinations on the same stem [45].

Ecologically, prickles play a crucial role in defense, deterring herbivores and potentially secreting antimicrobial compounds. They also aid in environmental adaptation, with density varying by climate; desert roses often exhibit denser prickles for water retention [46,47].

Molecular insights reveal that key genes such as TTG1, which regulates trichomes and prickles, and LOG, involved in cytokinin synthesis, are conserved across species. Quantitative trait locus (QTL) mapping has identified genomic regions, such as those on chromosome 3, linked to prickle density, facilitating marker-assisted breeding for prickle-free cultivars [41,42,43].

### 1.4. Background on the Almunia of Montagudo and Its Historical Significance

The extensive territory of the Castillejo de Monteagudo estate is located 5 km northeast of the city of Murcia, on the edges of the fertile plains irrigated by the Segura River (Figure 1). It is first mentioned in mid-12th-century Arabic sources as Qaṣr ibn Sa’d, clearly referring to Emir Abū ‘Abd Allāh Muḥammad b. Sa‘d b. Mardanīš al-Ŷuḏāmī, known in Christian chronicles as the Wolf King, who managed to make Murcia the capital of a vast taifa that spanned the eastern third of the Iberian Peninsula between 1147 and 1172 CE [48].

The fortified palace of Castillejo de Monteagudo has been well-known since its excavation a century ago and its subsequent publication by Torres Balbás. Recent surveys and excavations conducted in 2018, 2019, and 2023 have revealed that the Castillejo oversaw an extensive palatial estate. This estate included a large, official building that extended across the plain, as well as other residential structures. The estate featured an enclosing wall, which has not yet been located, encompassing cultivated lands (both irrigated and dry), wooded areas, and marshlands, along with significant hydraulic infrastructure.

The palace fell during the second Almohad campaign, which devastated the suburbs (*marbāḍāt*), orchards/estates (*basātīn*), and all the nearby lands of the plain (*basāiṭ*) and the hamlets (*qurà*) surrounding the middle area (*mūsaṭ*) of Murcia.

Ibn Ṣāḥib al-Ṣalā, who chronicled the Almohad campaigns against Murcia, confirms the existence of a rural estate (*almunia*) belonging to Ibn Mardanīš near the city, his capital. He refers twice to this country residence, identifying it once as *Ḥiṣn al-Faraŷ*. During the 1165 campaign, he describes how the Almohads advanced towards the outskirts (*finā’*) of Murcia, devastating both open spaces and built structures, including vineyards, cultivated plains (*basā’iṭ*), and large, well-constructed gardens (*basātīn*). These orchards, filled with fruit trees and myrtles, commonly featured buildings within them. The Almohad forces caused extensive destruction, cutting down and razing everything in their path [48].

In the 1171 campaign, the same source reports that the Almohads besieged Murcia, seized *Ḥiṣn al-Faraŷ*—described as Ibn Mardanīš’s pleasure estate (*mutanazzah*)—and devastated the suburbs (*marbāḍāt*), gardens (*basātīn*), cultivated plains, and nearby farmsteads (*qurà*) around the city center (*mūsaṭ*).

Half a century after Ibn Mardanīš’s death, the poet Ḥāzim al-Qarṭāŷannī mentions *Ḥiṣn al-Faraŷ* once more in his *Qasīda Maqṣūra*, alongside the place names Muntaqūd and *Qaṣr Ibn Saʿd* [48].

New structures were built over the ruins of some 12th-century buildings, dating to the 13th century. It is highly probable that these later constructions housed the court of Alfonso X during May and June of 1257, coinciding with the First Partition (*Primera Partición*), as evidenced by several royal charters issued by the king in Monteagudo [48]. Following the Castilian conquest of Murcia in 1243, the Monteagudo palace complex underwent significant architectural and functional transformations, reflecting the shifting political, social, and economic paradigms in the region. The almunia was subject to successive reuses and adaptations, affecting both residential structures and the surrounding gardens and agricultural spaces.

Recent archaeological work cautiously suggests that the original spaces, devastated in the 12th century, were transformed in the 13th century. The *Patio del Estanque* may have been converted into a covered garden with a trellis, indicated by a new circulation system and aligned postholes. Although no fossilized vegetation from the trellis remains, the spatial logic implies that the pond’s primary hydraulic function may have given way to a decorative garden, semi-open, adapted to the new residential needs of the late medieval period. The available evidence suggests that vine cultivation was prevalent on this property during the late 13th century but accompanied by extensive orchards and rain-fed crop fields. However, it remains unclear whether these vines were primarily grown for wine production, raisin production, or both [48].

In both the early 19th and 20th centuries, the area occupied by this palace appears to have been located at the boundary between irrigated farmland (such conditions may also have contributed to the survival of particularly resilient rose bushes) and the piedmont and slopes of the small *castillejo* of Monteagudo (Figure 1). During this period, these upland areas were covered with esparto grass and prickly pear (*Opuntia ficus-indica* (L.) Mill.) [32,33].

The Modern and Contemporary Era phase reveals continued rural use of the palace sector, culminating in the establishment of a traditional orchard house on the southern plot. This dwelling, documented in Ruiz de Alda’s 1928 aerial photograph (Figure 2), features a tripartite structure around a courtyard closed to the east. Remarkably, this courtyard coincides with the location of the old porticoed courtyard of the late Andalusian palace, suggesting uninterrupted occupational continuity from the Middle Ages to the 20th century. The tower, originating from the foundational phase (dated between the late taifa period and the Almoravid era), was adapted for domestic use by hollowing out its solid core and opening a door that was later sealed. This transformation persisted until the dwelling’s demolition in the 1960s.

These findings suggest continuous occupation of the palace space from the Andalusian era to contemporary times, indicating not a total rupture in occupation but rather a gradual reconfiguration of its uses and meanings. However, it appears that, at least in part, the estate underwent periods of abandonment, affecting both the drylands and the irrigated gardens, which gradually turned into marshy areas (*almarjales*).

Nevertheless, as this study would demonstrate, provided water remained available, it is highly likely that the rose bushes in the gardens could have survived for decades, reproducing vegetatively without human intervention—a phenomenon documented in recent cases of abandoned estates. This persistence would have enabled their recovery during successive phases of estate rehabilitation for agricultural use between the 14th and 18th centuries, with some rose bushes potentially surviving near inhabited areas or along the surrounding walls.

Throughout the 20th and 21st centuries (Figure 2), the complex underwent drastic transformations and terracing in the 1960s to facilitate irrigation on the slope of the “*castillejo*.” The cruciform garden of the “*castillejo”* itself was converted into a regulation pond for irrigating citrus trees planted on the terraces (Figure 2C). The lower area, where the palace and gardens under excavation are located, was also affected by citrus planting. Approximately twenty years later, the initial process of reevaluating the archaeological site put an end to the irrigation of the slope.

### 1.5. Hypothesis and Objectives of the Study

Given the archaeological significance of rose cultivation in Islamic gardens and the challenges inherent in identifying fragmentary botanical remains, we hypothesize that the rose prickles recovered from post-medieval layers at Almunia de Monteagudo represent specific cultivated taxa that can be distinguished through morphometric analysis. The present study aims to: (1) document and characterize the rose prickle assemblage from the Islamic Almunia de Monteagudo, Murcia; (2) develop a Bayesian probabilistic framework for species identification based on comparative morphometric analysis of modern rose prickle assemblages; and (3) determine the most probable botanical sources of the archaeological specimens to enhance understanding of rose cultivation practices in post-medieval Iberian gardens. This represents the first application of Bayesian inference to archaeological rose identification, providing a quantitative approach to taxonomic assignment that addresses the morphological variability inherent in these botanical remains.

## 2. Results

### Description of the Findings, Including the Condition and Distribution of the Rose Prickles

Five prickles with distinct and characteristic morphologies were recovered, and two were radiocarbon dated. This discovery prompted the use of a Bayesian approach for their tentative identification on probabilistic grounds. The Bayesian approach enabled us to probabilistically link the archaeological prickles to the various hypothetical botanical sources (Table 2). The analysis emphasized information derived not only from the proportion of types but also from available information from medieval literature.

Table 2 presents a Bayesian probabilistic analysis designed to identify the botanical sources of the two rose prickles recovered from post-medieval contexts at Almunia del Castillejo in Murcia, Spain. The analytical framework is particularly sophisticated in its treatment of uncertainty, encompassing the two distinct morphological prickle types and employing two alternative prior probability structures to assess the robustness of source attributions.

The temporal context of these finds is significant: MAS radiocarbon dating places both prickles (Figure 3) within a broad post-medieval window spanning from the 17th through late 19th centuries. This chronological framework comprehends a period of substantial horticultural development in the region, during which both indigenous wild roses and introduced ornamental cultivars would have been present in the landscape. The prior probabilities in Scenario A (Table 2) reflect this complexity, drawing not only from medieval documentary sources but also from later historical records and biogeographical data concerning both wild and cultivated roses in the surrounding area.

The morphological distinction between the two prickle types provides crucial functional and taxonomic information (Figure 3). Type 1 prickles (186 ± 26 BP 68% probability) exhibit a hooked morphology, characteristic of climbing or rambling roses, which use such structures to anchor themselves while ascending through vegetation or along architectural supports. This type was recovered from samples of 2019 and 2023 excavations. Type 2 prickles (dated 91 ± 26 BP with 68.2% probability), by contrast, display a straight, sloped form typically associated with shrub roses that maintain an upright, self-supporting growth habit. This morphological dichotomy strongly suggests that the archaeological assemblage derives from two distinct rose taxa with fundamentally different growth strategies. This second type was exclusively recovered from 2023 excavation. Furthermore, the two types of identified prickles have distinct and different botanical origins, although both types can coincide in stems of one wild taxon (*R. agrestis*) and two cultivated taxa (*R.* × *alba*, *R.* × *damascena*).

The Bayesian analysis reveals markedly different source attributions for each prickle type, with the pattern remaining consistent across both prior probability scenarios despite substantial differences in their underlying assumptions. For Type 1 prickles, *Rosa moschata* emerges as the most probable source under the historically informed priors of Scenario A, achieving a posterior probability of 0.54. This musk rose, renowned for its vigorous climbing habit and intensely fragrant flowers, was extensively cultivated in gardens throughout the Mediterranean basin during the medieval and post-medieval period. Its prominence in the analysis aligns well with both the hooked morphology of Type 1 prickles and the horticultural traditions of southern Spain.

However, when uniform priors are applied in Scenario B (Table 2), *Rosa canina* becomes the dominant candidate for Type 1 prickles, with a posterior probability of 0.50 compared to 0.28 for *R. moschata*. This dog rose is a widespread wild species native to the region, characterized by a vigorous scrambling growth habit that employs hooked prickles for mechanical support. The substantial probability assigned to *R. canina* under uniform priors reflects its natural abundance and morphological compatibility with Type 1 specimens, though its probability decreases notably when historical and biogeographical constraints are incorporated into the prior distribution. *Rosa agrestis*, another wild species, shows moderate probabilities across both scenarios, suggesting it cannot be entirely excluded as a potential source.

The attribution pattern for Type 2 prickles presents a striking contrast, with overwhelming support for two closely related yellow-flowered species. *Rosa lutea* var. *persiana*, commonly known as Persian Yellow rose, achieves posterior probabilities of 0.52 under Scenario A and 0.44 under Scenario B, making it the most probable source for these straight-prickled specimens across both analytical frameworks. *Rosa foetida*, the Austrian Briar, follows closely with probabilities of 0.30 and 0.25, respectively. Both species are characterized by upright shrub growth habits and straight prickle morphology, perfectly consistent with the Type 2 archaeological specimens. This aligns with references to yellow roses by Columella in Roman times [50,51] and in medieval times by Abū’l-Jayr al-Ishbīlī, Al-Ṭighnarī, and Ibn al-ʽAwwām of the yellow roses from Alexandria (Egypt) and by Ibn Baṣṣāl and Ibn al-ʽAwwām of the roses of yellow daffodil color [24,25,26,27,28,29,30].

The robustness of *R. lutea* var. *persiana* and *R. foetida* as candidate sources across both prior probability structures is particularly noteworthy. Unlike the Type 1 attributions, which show substantial sensitivity to prior assumptions, the dominance of these yellow roses for Type 2 prickles remains stable whether historically informed or uniform priors are employed. This consistency strengthens the interpretation that these cultivated ornamental species were genuinely present at the site during the 17th through 19th centuries.

Several taxa can be effectively eliminated from consideration based on negligible posterior probabilities across both prickle types and both prior scenarios. *Rosa bicolor*, *Rosa gallica*, the Tea rose group, and *Rubus ulmifolius* all show probabilities approaching zero, indicating that the morphological characteristics of the archaeological prickles are incompatible with these species. Similarly, the various hybrid roses, including *R.* × *alba*, *R.* × *bifera*, *R.* × *centifolia*, and *R.* × *damascena*, show minimal probability despite receiving elevated priors in Scenario A (Table 2) based on their historical cultivation in the region.

The contrasting patterns between prior scenarios illuminate the epistemological challenges inherent in archaeological plant identification. The historically informed priors of Scenario A privilege cultivated ornamental species that documentary and biogeographical evidence suggests were present, resulting in strong support for *R. moschata* over the wild *R. canina* for Type 1 prickles. The uniform priors of Scenario B adopted a more agnostic stance, allowing the wild species to achieve prominence based on likelihood alone. The fact that *R. moschata* maintains substantial probability even under uniform priors, while *R. canina* shows markedly reduced probability under informed priors, suggests that both species warrant thoughtful consideration as potential sources, with the true attribution depending on the specific depositional context and landscape setting from which each specimen was derived.

This analysis provides quantitative evidence for the presence of both wild climbing roses and cultivated ornamental shrub roses at Almunia del Castillejo during the post-medieval period. The probable identification of *R. lutea* var. *persiana* or *R. foetida* is particularly significant, as these species represent intentional horticultural introductions requiring deliberate cultivation and maintenance. Their presence, alongside evidence for *R. moschata*, another prized garden plant, suggests continuity of ornamental gardening traditions at the site extending well beyond the medieval Islamic period for which such almunias are best known. In the case of *R. moschata*, it will imply a persistence of this rose type from the Andalusian garden since the time of the estate’s destruction, according to the extensive use of this species in Southern Spain as documented by writers of al-Andalus and travelers. Furthermore, the Winter Rose of New Carthage (modern Cartagena), a locality situated 50 km away from Monteagudo, Spain, should be *Rosa moschata* Herrm. It “Blooms throughout winter”, as mentioned by Pliny the Elder, 1st century CE [52]. This is consistent with the existence of fragrant white roses cultivated in North Africa prior to the Roman conquest, linked to the Phoenicians and Carthaginians. The fact that Cartagena was a Carthaginian foundation on a coast with a Phoenician presence dating back several centuries supports this. Additionally, the roses cultivated in Posidonia or Paestum, south of Salerno, from the 4th century BCE, were remontant hybrids of *R. gallica* and *R. moschata*. Furthermore, *Rosa moschata* (Figure 4A,C) continued to be cultivated for centuries in Tunisia, the territory of ancient Carthage, according to Desfontaines [53].

The potential presence of wild roses such as *R. canina* or *R. agrestis* would indicate either spontaneous colonization of the site or deliberate exploitation of native species within the managed landscape. The morphological complementarity between the two prickle types and their respective most probable sources strengthens confidence in these attributions and demonstrates the utility of Bayesian frameworks for handling uncertainty in archaeobotanical interpretation.

## 3. Discussion

### 3.1. Interpretation of the Results in the Context of Medieval Garden Practices

Three rose prickles similar to the Type 2 were recovered by Matilla-Seiquer [56] from Tell Qara Quzaq in the Euphrates area of north Syria in levels of Middle Bronze, silos 27 (2 prickles) and 78b (1 prickle).

In medieval Western Muslim contexts, rose bushes are ornamental plants cultivated for visual and olfactory delight, recommended for planting along garden sides, walkways, or near pavilions. They have numerous uses in medicine and produce rose water [19,20,21,22,23,24,25,26]. The Cordova Calendar, from the Umayyad Caliphate under al-Hakam II (961–976 CE), lists roses amongst around one hundred plants, including utilitarian and ornamental species like chamomile, iris, jasmine, myrtle, narcissus, and violet [23].

The present results suggest the continuity in the cultivation of fragrant *Rosa moschata* in the Mediterranean, originating from Iran and Afghanistan, and the ancestor of *Rosa* × *bifera*, the rose of Paestum, through successive hybridizations with *R. gallica* L. from Europe, Western Asia, and the Caucasus, and *R. webbiana* Wall. ex Royle (=*R. fedtschenkoana* Regel) from Central Asia. In addition to providing strong evidence of the cultivation of yellow-flowered roses from Western Asia, Central Asia, and the Caucasus, this finding suggests that these yellow roses were likely the same varieties mentioned by Columella in the 1st century CE [50,51].

### 3.2. Possible Implications for Understanding the Use and Significance of Roses in Medieval Muslim Gardens

Rose remains and particularly fruits and prickles recovered from archaeological strata may deeply help to clarify our understanding of the role of roses in past times. Gardens of al-Andalus underwent in Post-Medieval times cycles of abandonment and revival in Post-Medieval times. During periods of decadence, gardens may have been abandoned and later reconstructed.

### 3.3. Limitations of the Study and Potential Avenues for Further Research

Limited sample size, reduced to a pair of single prickles, hinders a comprehensive analysis of intra-sample or within-site heterogeneity. This study paves the way for further research on the medieval diversity of cultivated roses, emphasizing the taxonomic value of rose prickles due to their pronounced polymorphism. In archaeology, it introduces a novel framework for valuing these previously understudied plant remains, which are crucial for reconstructing the evolution of gardening and horticultural preferences over the centuries. Our study is based on a comprehensive comparison with botanical roses and ancient varieties we cultivated, which are medieval according to consulted sources. Future research could expand the number of varieties examined to provide a more robust understanding. Additionally, while our comparison between archaeological and modern prickles was conducted empirically using digital images, the study would have benefited from the combined use of image analysis techniques and deep learning.

## 4. Materials and Methods

### 4.1. Description of the Site and Its Location

The estate known as Castillejo de Monteagudo is located 5 km northeast of the city of Murcia. It was a state-owned property during the Muslim period, covering over 111 hectares, excluding barren lands. The estate featured a mix of irrigated and dry farming areas, along with forested regions and marshlands. As was typical for such estates, it served a triple function: economic, as an agricultural and livestock operation; residential, with various palatial buildings suitable for seasonal habitation; and ceremonial, as a venue for audiences and meetings with ambassadors and dignitaries.

The three excavation campaigns (2018, 2019, and 2023) were conducted in the flat garden area extending west of the fortified palace. These excavations uncovered a previously unknown palatial area, featuring several residences of varying sizes organized around a large cruciform garden. Similar to the *alcázar* of Madīnat al-Zahrāʾ (10th century CE), the main palace of the Murcian estate (12th century CE), fronted by a portico and a pool, opens to the landscape, emulating the design of the upper and lower gardens of the aforementioned Cordovan alcazar (Figure 5).

In the case of the West sample (UE 22003), excavated in 2019, the rose thorn (Prickle Type 1, *Rosa moschata* Herrm.), dated 186 ± 26 BP (calibrated date 1720–1820 AD), was recovered from a stratum located approximately 60 cm below the current surface in an area that had already been transformed into a cultivation space during the 13th-century reforms. These modifications involved the partial dismantling of the cruciform garden and its conversion into a trellised garden area. Although the 20th-century orchard house is located nearby but not directly above the sample, it cannot be ruled out that this space was part of its garden surroundings. What does seem clear is the continuity of garden-related uses from the 13th century to at least the modern era, which would explain the modern or contemporary chronology obtained for this sample as a result of the functional persistence of the area as a green space, rather than an anomalous intrusion. Therefore, the modern chronology obtained through radiocarbon dating for this thorn is explained by a contemporary vegetal intrusion into sediment altered by agricultural practices, in a context where the depth of tilled earth favors the vertical mixing of recent organic materials.

Regarding the East sample (UE 91003), Prickle Type 2, *Rosa lutea var. persiana* Lemaire ex Duchartre, dated 91 ± 26 BP (calibrated date 1800–1930 AD), it was recovered from a survey conducted in 2023 in the reserved gardens sector of the almunia, within a closed deposit associated with the expansion following the initial destruction of the complex in 1165. Although the stratum was completely sealed, it was not hermetic. The modern chronology obtained for this thorn can be explained by bioturbation processes, such as the action of insects or other organisms, which may have introduced recent organic material through small fissures in the pavements sealing the deposit. The rose prickle assemblage (n = 4) (3 of type 2 and 1 of type 1) occurred within a diverse botanical context comprising remains from 16 plant taxa (total n = 61 identifiable specimens). The assemblage was numerically dominated by ruderals and herbaceous species—*Commicarpus plumbagineus* (n = 16), *Medicago polymorpha* (n = 10), *Malva parviflora* (n = 1), and *Eragrostis papposa* (n = 5)—alongside members of Amaranthaceae (*Amaranthus albus*, *Chenopodiastrum murale*), Brassicaceae (*Brassica fruticulosa* subsp. *cossoniana*), and cultivated species including *Borago officinalis* (n = 6), *Spinacia oleracea*, and *Ficus carica*. Ornamental taxa were represented by *Narcissus tazetta* (n = 2). Additional *Rosa* sp. bud (n = 1) was present but lacked diagnostic features for species-level identification.

This biological intrusion does not necessarily imply significant alteration of the archaeological context as a whole.

### 4.2. Excavation Methods

In the 2018 and 2019 archaeological excavation campaigns at Almunia del Castillejo de Monteagudo, a total of 25 samples were collected, with 14 from 2018 and 11 from 2019. This study aimed to reconstruct the vegetation of this palatial estate, including ornamental, productive, and adventitious plants. The recovery of plant material was systematically planned and involved analyzing various archaeological levels. Each stratigraphic unit was sampled uniformly, allowing for the comparison of archaeobotanical remains with different anthropic activities. This method is highly effective for extensive excavations.

A constant volume of 20 L of soil per stratigraphic unit was processed during excavation. Utilizing the site’s abundant water supply, a simple flotation method was employed for recovering plant remains. Between 60 and 80 L of water per 20 L of soil were used, with sediment and water mixed at a ratio of 5–7 L of sediment per 20 L of water. The mixture was filtered through sieves with mesh sizes of 2 mm, 1 mm, and 0.5 mm. No detergents or flocculants were added to facilitate flotation. The samples were analyzed at the Faculty of Biology, University of Murcia. The isolation of individual remains from the assemblage was conducted under a Olympus (Tokyo, Japan) SZ binocular microscope, ranging from 10 to 40× magnification, and subsequently stored in Eppendorf polymer tubes of varying sizes.

### 4.3. Radiocarbon Dating

The biotic assemblages recovered from Almunia de Monteagudo include two desiccated samples of short-lived botanical material, specifically rose thorns or prickles (*Rosa* sp.). Radiocarbon dating was conducted at the Royal Institute for Cultural Heritage, Brussels, under laboratory identification codes RICH (current designation). All radiocarbon determinations were calibrated using the radiocarbon calibration program CALIB 8.2 [57,58] (Table 3).

In parallel, a Bayesian analysis of radiocarbon dates was performed using OxCal, a program developed by Bronk Ramsey in 1994 and subsequently modified in 2009 [59,60]. The analysis yielded 2σ probability ranges (corresponding to 95.4% confidence intervals) for both samples examined.

For sample UE22003, the 2σ probability range presents a maximum surface area for the curve between 1726 and 1811 AD. However, the complete probability distribution extends from 1626 to 1918 AD, with potential extension beyond these boundaries.

Similarly, sample UE91003 exhibits a 2σ probability range with maximum surface area between 1807 and 1924 AD, while the full probability distribution spans from 1689 to 1924 AD, again with extension beyond these limits.

Overall, this underlines the difficulties of this estimate but replicates result of the former method.

### 4.4. Identification Process of the Rose Prickles

The starting hypothesis for the identification of the prickles recovered is that they belong to a species of rose bush that would grow under the environmental conditions prevailing in Murcia from the medieval *bustān* to the modern period, and that could be a species cultivated for ornamental purposes or, although unlikely considering altitude and climate of the Monteagudo zone, a rose that grew spontaneously on the margins of the irrigation ditches within the farm or, alternatively, a species of blackberry typical of the area (*Rubus ulmifolius*) (Figure 6) [61,62]. For this purpose, we have considered the medieval Andalusian literature [26] with references to this type of plant and the studies on the flora of the territory where the Almunia de Monteagudo is located, the Lower Segura River area [61,62].

According to the data of Ríos et al. [61] and Anthos (2024) [62], we must consider extremely improbable the presence of wild roses around the *almunia,* except for the occasional presence of *R. agrestis*, and, at most, there could be *Rosa canina* as rootstock of other roses, but also unlikely, this limits us to the cultivated roses known from the Medieval period to the 19th century, or *Rubus ulmifolius*. Consequently, we prepared a training set of rose scions with thorns from the species that are likely to be found in medieval and post-medieval contexts in southern Spain, both wild and cultivated (Figure 6), excluding modern cultivars obtained during the 20th and 21st centuries.

Ninety-one stem fragments and seven leaf rachis fragments from sixteen *Rosa* taxa (species and varieties), and two stem fragments and three leaf rachis fragments from one sample of *Rubus ulmifolius* were analyzed as modern reference materials (Appendix A). Further, four stem fragments, carbonized, from villa B in Oplontis (Italy), of the Applied Research Laboratory in Pompeii, preserved among the ashes of the Vesuvius eruption in 79 CE [63] were included. The fragments were photographed using a macro photography setup consisting of a Nikon D80 camera, a Micro Nikkor 60 mm f/2.8 G ED AF lens, and a Nissin Macro M 18 ring flash. Each fragment from the sample was carefully arranged on graph paper to facilitate precise imaging. On the digital images of each fragment, the three types of prickles (T1: short, hooked with a wide base, 2.5 mm × 3 mm; T2: slightly curved, thin, 5 mm × 2 mm; and TO: others) were counted, and the fragment length was measured, ranging between 3 and 9 cm. The average number of prickles of each type per taxon and sample type (leaves or stems) was estimated, along with their standard deviation (Appendix A).

The estimated number of prickles per plant, calculated based on the observed prickle density per cm in the studied training set and assuming each plant had twenty 1.5 m long stems, resulted in a range between 500 and 80,000 prickles.

### 4.5. Bayesian Hypothesis Testing: Explanation of the Approach and Its Application

This method was previously utilized for archaeological seed interpretation [64,65] and for modeling ancient areas of date palm through multiple evidence integration by Rivera et al. (2020) [66,67]. When the Bayesian approach is used to determine the conditional probability of an archaeological seed, or prickle in this case, belonging to a specific taxon Θ*_i_*. We sought to answer: What is the likelihood that an archaeological or sample is assigned to Θ*_i_* given its distinctive characteristics and the average proportion of such a type in each one of the considered taxa value *x_j_* and/or its relative proportion considering the abundance of prickles per rose bush *y_j_*?

Drawing upon a dataset of 97 comparison samples, from 14 rose taxa and 1 *Rubus* taxon, where taxonomic identity is known a priori, from previous morphological studies of the whole plant, we constructed a discrete joint probability function *p*(*X*,Θ). This function assigns a posterior probability value to each prickle type within every *Rosa* taxon, considering their relative proportions.

Bayes’ rule (1) was employed to approximate the solution.
(1)$$p\left(\theta |x\right)=p\left(x|\theta \right)p(\theta )/p(x)$$ where *p*(*θ*|*x*) is the posterior probability distribution for the parameter *θ*, in our case the species of rose, given a single observed value of the variable *X* = *x_j_*, in our case the type of prickle (T1, T2 or TO).

When considering the Bayes’ rule in terms of individual probabilities Formula (1) can be read as (2).
(2)$$posterior\;probability=\frac{likelihood\;x\;prior\;probability}{marginal\;likelihood}$$

Given a value for the data, for instance *X* = *x*_2_ and a specific value for the parameter *θ* (rose taxa), such as, *θ* = *θ*_3_, we get (3).
(3)$$p\left(\theta_{3}|x_{2}\right)=p\left(x_{2}|\theta_{3}\right)p(\theta_{3})/p(x_{2})$$

In (3), both likelihood *p*(*x*_2_|*θ*_3_) and marginal likelihood *p*(*x*_2_) (4) can be calculated based on the joint distribution derived from the training samples.
(4)$$p(x_{2})=\sum_{i=1}^{n}p\left(x_{2}|\theta_{i}\right)\times p(\theta_{i})$$

The prior probability *p*(*θ_i_*) can also be computed as the sum of probabilities of this taxon given the distribution of all *x* values, solely from the sample data, but this requires careful sampling strategies to avoid experimental biases. Nonetheless, the prior’s nature allows for the integration of data on regional prevalence of different taxa from other established sources of evidence. In this study, we adopted different approaches: first, a neutral position, if the available evidence does not allow for establishing such a distinction; second, prior probability is calculated based on the analysis of literature reviewed by Rivera et al. (in press) (Appendix A). In this study, we approached prior probabilities for each taxon, considering the medieval and early Renaissance literature concerning Western Muslim gardens and agriculture, along with the available botanical information from the area. The estimated prior probabilities are represented in Table 2.

## 5. Conclusions

Morphometric and radiocarbon analyses conclusively identify the rose prickles recovered from Monteagudo Almunia as *Rosa moschata* and *Rosa foetida* (or *R. lutea* var. *persiana*). AMS dating establishes a post-medieval chronology, spanning the 17th to 19th centuries, demonstrating the critical importance of direct dating rather than relying on stratigraphic context alone.

These findings underscore the necessity of systematic collection protocols for botanical macroremains at medieval excavation sites. Beyond the conventional focus on seeds and fruits, identifiable vegetative structures—including prickles, thorns, wood fragments, and bark—warrant equal attention. Such materials offer complementary taxonomic resolution and enable direct chronological assessment through radiocarbon dating.

Future archaeobotanical research would benefit from integrating computational approaches for morphometric identification. Bayesian statistical frameworks and deep learning algorithms for image analysis can enhance taxonomic precision and enable systematic comparison across assemblages. When coupled with routine AMS dating protocols for botanical materials, these methods provide robust chronological control essential for reconstructing landscape evolution and agricultural practices over millennial timescales.

The Monteagudo case study demonstrates that apparent medieval contexts may contain intrusive later materials, reinforcing the imperative for independent chronological verification of botanical evidence in long-occupation sites.

## Figures and Tables

**Figure 1 plants-14-03709-f001:**
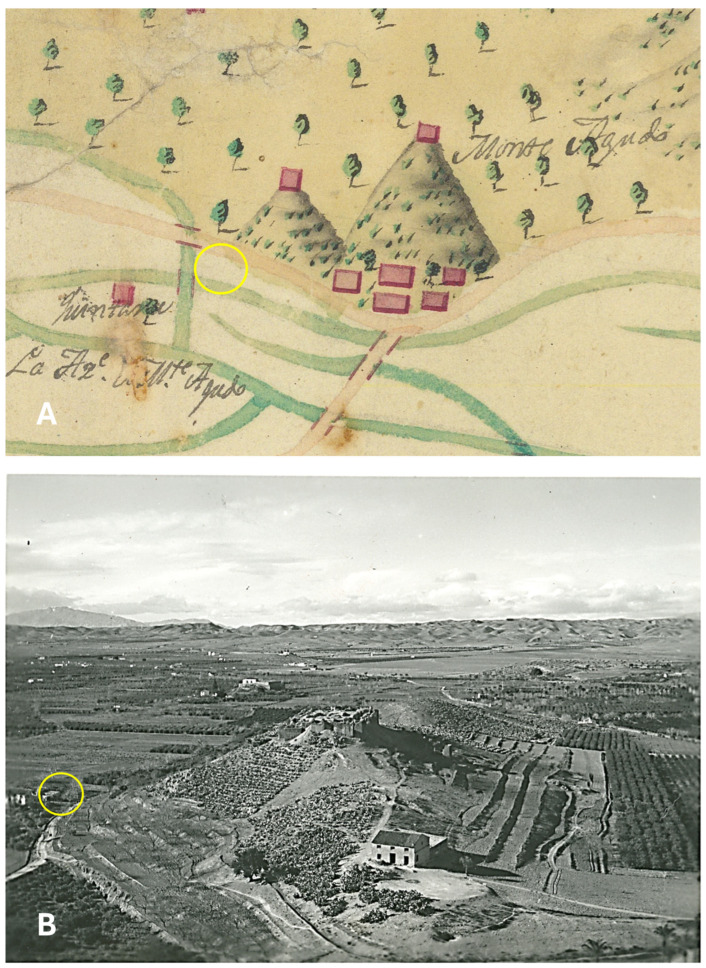
Approximate Location of the Study Site in the Monteagudo Area (Murcia, Spain). (**A**) An 1809 military map illustrating the historical irrigation channel network and its potential use for flooding the Huerta de Murcia, providing context for traditional water management practices in the region. The manuscript original labels can be read as: “Quintana” on the left side, that would likely refer to a farmhouse, “Monte Agudo”—on the right side near the top, to present Monteagudo castle and “La az(arb)e de Monte Agudo”—at the bottom left, referring to the channel (azarbe) of Monte Agudo. The azarbes are drainage systems in the irrigation complex of Murcia. (**B**) A 1925 photograph taken from Monteagudo Castle, capturing the “Castillejo” at the hill’s summit. In (**A**) and (**B**) the yellow circle is marking the approximate location of the excavated area for reference. Images A courtesy of the Julio Navarro Archive B, Cliché Avilés. Archivo General de la Región de Murcia, FOT_POS, 211/173.

**Figure 2 plants-14-03709-f002:**
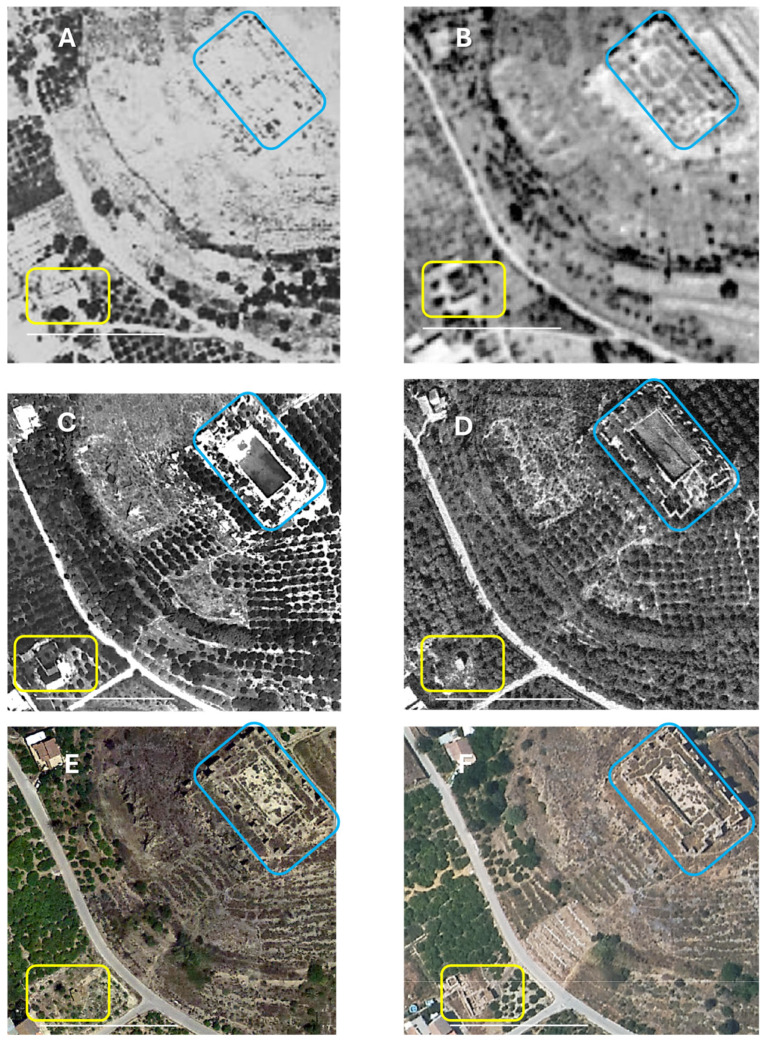
Location and Historical Evolution of the Study Site in Monteagudo (Murcia, Spain). The images illustrate the spatial distribution of natural vegetation, agricultural plantations, and non-vegetated open areas within the study site. The series also depicts the land-use changes and landscape transformation that have occurred over the last century, providing insight into the dynamic interplay between human activity and ecological systems in the region. (**A**) Ruiz De Alda in 1928–30; (**B**) Americano in 1957; (**C**) Geofasa in 1969; (**D**) Interministerial in 1977; (**E**) Pnoa in 2016; (**F**) Pnoa in 2022. The “Castillejo” is highlighted in blue, and the area of the palace and gardens from where the rose prickles were recovered are highlighted in yellow. Scale = 100 m. All images from Confederación Hidrográfica del Segura, 2025 [49], processed by Diego Rivera.

**Figure 3 plants-14-03709-f003:**
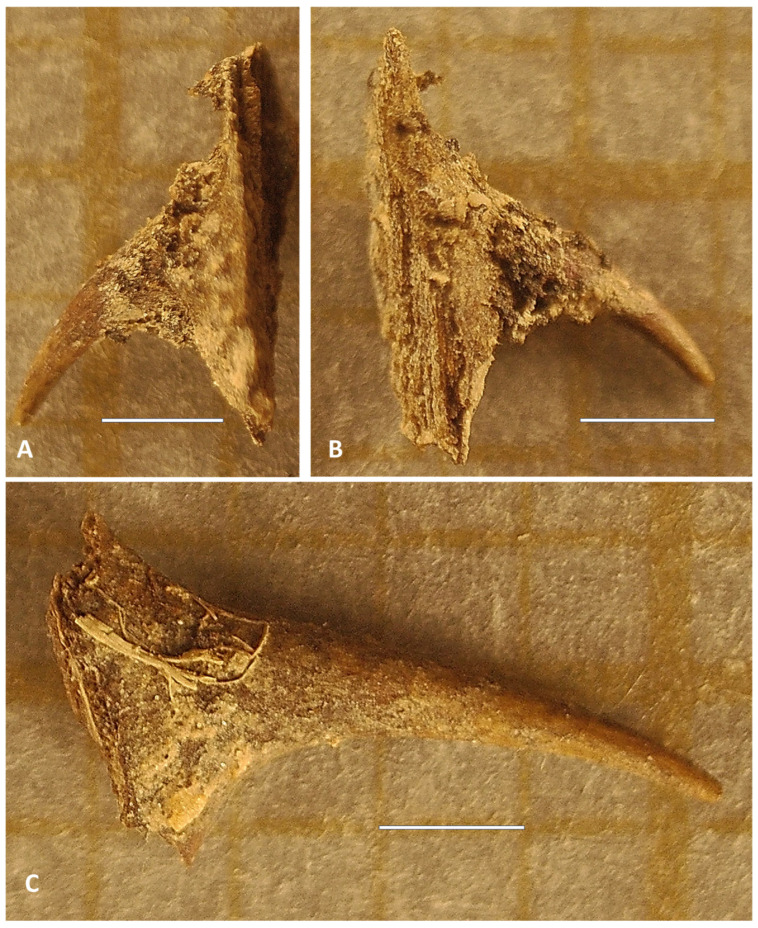
Archaeobotanical rose remains recovered from the Almunia del Castillejo (Monteagudo, Murcia, Spain) and radiocarbon dated. (**A**,**B**) Short, hooked prickle with wide base 2.5 × 3 mm, Prickle Type 1, *Rosa* sp., 186 ± 26 BP 68.2% probability: the West sample, excavated in 2019. (**C**) Thin, slightly curved, medium-sized, narrow-based prickle, 4 × 2.5 mm, Prickle Type 2, *Rosa* sp., 91 ± 26 BP 68.2% probability: the East samples excavated in 2023. Scale = 1 mm. Images (**A**–**C**) by Javier Valera.

**Figure 4 plants-14-03709-f004:**
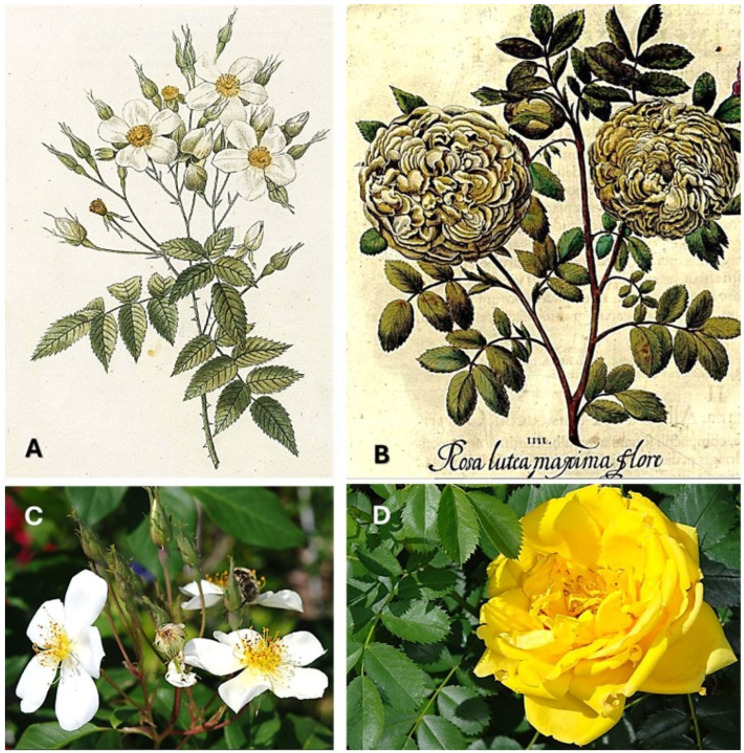
Probable rose species identified as botanical sources for the analyzed rose prickles. (**A**,**C**) *Rosa moschata* Herrm. (**B**,**D**) *Rosa lutea var. persiana* Lem. [syn. *Rosa foetida* Herrm. var. *persiana* (Lemaire ex Duchartre) Rehder]. Note in (**B**) the original pre-Linnean polynomial name *Rosa lutea maxima flore* by Bessler. Historical botanical illustrations: (**A**) Redouté (1828, Vol. 1) [54]; (**B**) Bessler (1613) [55]. Contemporary photographs: (**C**,**D**) Diego Rivera and Concepción Obón.

**Figure 5 plants-14-03709-f005:**
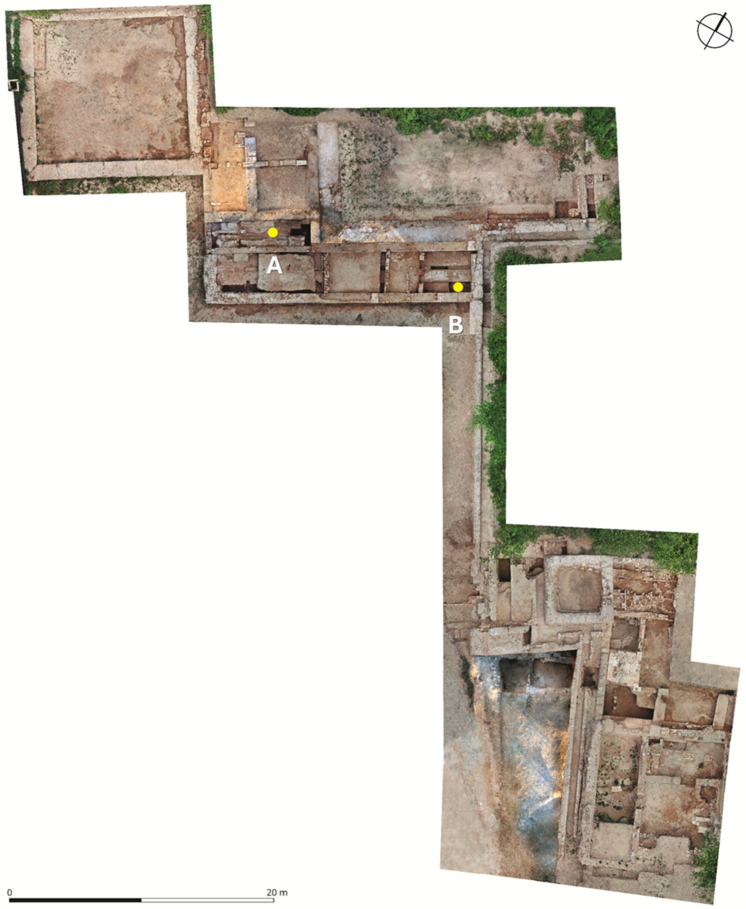
Excavated Area Showcasing Sample Locations and *Rosa* Taxa Identification. (**A**) West sample site, where Prickle Type 1 was recovered and identified as *Rosa moschata* (with ~70% probability). Radiocarbon dating places the specimen at 186 ± 26 BP (with 68.2% probability). (**B**) East sample site, where Prickle Type 2 was recovered and identified as *Rosa lutea* var. *persiana* Lem. (with 54% probability). Radiocarbon dating dates this specimen to 91 ± 26 BP (with 68.2% probability). Image credit: Jose Javier Martínez (used with permission).

**Figure 6 plants-14-03709-f006:**
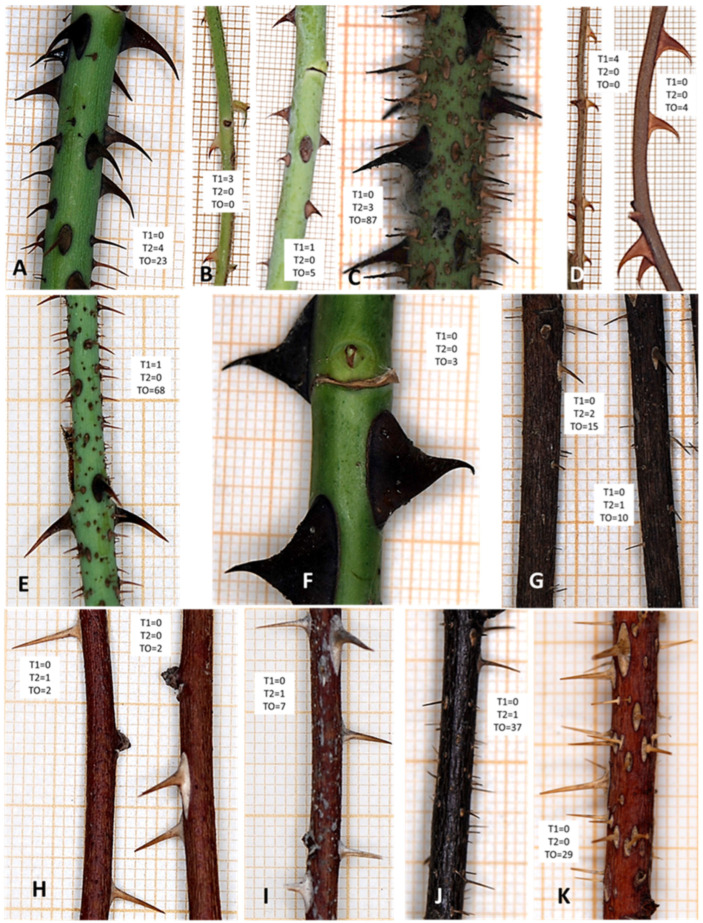
Examples of *Rosa* Taxa from the Training Dataset, Showcasing Variability in Prickle Morphology. The images highlight representative specimens of different *Rosa* taxa, illustrating the diverse forms and arrangements of prickles used as diagnostic features in the training set. (**A**) *Rosa* × *alba* “Felicité Parmentier”. (**B**) *Rosa moschata*, left, leaf rachis, right stem. (**C**) *Rosa* × *centifolia* “Muscosa”. (**D**) *Rosa canina*, left, leaf rachis, right, stem. (**E**) *Rosa* × *damascena* “Kazanlik”. (**F**) Rosa Hybrid Tea group. (**G**) *Rosa gallica* “Officinalis”. (**H**) *Rosa foetida* (Mesones). (**I**) *Rosa lutea* var. *persiana* (Molina de Segura). (**J**) *Rosa hemisphaerica* (Molina de Segura). (**K**) *Rosa bicolor*. Scale = 1 mm. Abbreviations: T1, type 1, hooked prickle with wide base 2.5 × 3 mm. T2, type 2, thin, slightly curved, medium-sized, narrow-based prickle, 4 × 2.5 mm. TO, other types in reason of size or shape. Images by Diego Rivera.

**Table 1 plants-14-03709-t001:** Diversity of roses (*Rosa* L.) known according to Western Muslim authors, 1000–1200 CE.

Likely Species	Flower Type	Characteristics	Authors
*R. bicolor* Jacq.	Purple & yellow	Roses yellow on the outside and blue (Purple) inside	Ibn al-ʽAwwām
*R.* × *centifolia* L.	Red	Double red roses	Ibn Baṣṣāl and Ibn al-ʽAwwām
*R. gallica* L.	Red	Bright red roses	Abū’l-Jayr al-Ishbīlī
*R. pulverulenta* M.Bieb. (syn. *R. sicula* Tratt.)	Red	Magian roses with five petals, red, found in the East and al-Sham	Al-Ṭighnarī and Ibn al-ʽAwwām
*R.* × *alba* L. or a white *R.* × *centifolia*	White	Intensely white or camphorated roses with more than a hundred petals	Abū’l-Jayr al-Ishbīlī, Ibn Baṣṣāl, and Ibn al-ʽAwwām
*R. moschata* Herrm.	White	Chinese roses (*Ward al-ṣīnī*)	Ibn Luyūn, Abū’l-Jayr al-Ishbīlī and Ibn al-ʽAwwām
*R. sempervirens* L.	White	White orchard rose bush, smaller with narrower leaves and smaller flowers	Abū’l-Jayr al-Ishbīlī
*Rosa* × *alba* L.	White	White roses from the land of the Slavs and Persia	Abū’l-Jayr al-Ishbīlī
*R.* × *damascena* Herrm.	White & red	Double roses of superior quality, white with red tinges	Abū’l-Jayr Ishbīlī, Ibn Baṣṣāl, Al-Ṭighnarī, and Ibn al-ʽAwwām
*R. canina* L., and other species	Wild	Wild mountain roses	Abū’l-Jayr al-Ishbīlī
*R. canina* L.	Wild	Wild roses (*Nisrin*), onto which cultivated roses are grafted	Abū’l-Jayr al-Ishbīlī
*R. lutea* var. *persiana* Lem. and/or *R. foetida* Herrm.	Yellow	Yellow roses from Alexandria	Abū’l-Jayr al-Ishbīlī, Al-Ṭighnarī, and Ibn al-ʽAwwām
*R. rapinii* Boiss. & Balansa or *R. hemisphaerica* Herrm.	Yellow	Roses the color of yellow daffodils	Ibn Baṣṣāl and Ibn al-ʽAwwām
Artificial (Ibn al-ʽAwwām)	Blue	Blue roses in various shades	Abū’l-Jayr al-Ishbīlī, Al-Ṭighnarī, and Ibn al-ʽAwwām
Unknown	Wild	Wild white mountain roses	Al-Ṭighnarī
Unknown	Wild	Wild white to red mountain roses, with twenty to thirty petals	Al-Ṭighnarī
Unknown	Blue	Dark roses (*aswad*), the color of violets, from Syria and Lebanon	Abū’l-Jayr al-Ishbīlī, Ibn Baṣṣāl, and Ibn al-ʽAwwām
Unknown	Blue & red	Roses red on the outside and blue inside	Al-Ṭighnarī and Ibn al-ʽAwwām
Unknown	Blue & yellow	Roses with yellow on the inside and blue on the outside, found in Baghdad and Tripoli of al-Sham	Al-Ṭighnarī and Ibn al-ʽAwwām

Note: Data summarized from [20,21,22,23,24,25,26].

**Table 2 plants-14-03709-t002:** Bayesian analysis of probable botanical sources for the archaeobotanical rose (*Rosa* L.) prickles from Almunia del Castillejo (Monteagudo, Murcia, Spain). Estimates are based on the proportional abundance of prickle Types 1 and 2 across different taxa and two alternative prior probabilities. (A) Prior probabilities estimated on the basis of historical and biogeographical constraints. (B) Prior probabilities assumed uniform.

	(A) Prior Under Historical and Biogeographical Constraints			(B) Prior Uniform		
Taxa	Prior Probability	p (Taxon Given T1)	p (Taxon Given T2)	Prior Probability	p (Taxon Given T1)	p (Taxon Given T2)
*R. agrestis* Savi	0.11	0.23	0.04	0.06	0.12	0.03
*R. bicolor* Jacq.	0.04	0.00	0.00	0.06	0.00	0.00
*R. canina* L.	0.02	0.18	0.00	0.06	0.50	0.00
*R. foetida* Herrm.	0.11	0.00	0.30	0.06	0.00	0.25
*R. gallica* L.	0.02	0.00	0.00	0.06	0.00	0.02
*R. hemisphaerica* Herrm.	0.05	0.00	0.02	0.06	0.00	0.03
*R. lutea* var. *persiana* Lem.	0.11	0.00	0.52	0.06	0.00	0.44
*R. moschata* Herrm.	0.11	0.54	0.00	0.06	0.28	0.00
*R. rubiginosa* L.	0.01	0.00	0.01	0.06	0.00	0.08
*R. sempervirens* Savi	0.01	0.01	0.00	0.06	0.08	0.00
*R.* sp. Tea	0	0.00	0.00	0.06	0.00	0.00
*R.* × *alba* L.	0.05	0.01	0.02	0.06	0.02	0.04
*R.* × *bifera* (Poir.) Pers.	0.05	0.00	0.03	0.06	0.00	0.05
*R.* × *centifolia* L.	0.11	0.00	0.04	0.06	0.00	0.04
*R.* × *damascena* Herrm.	0.11	0.02	0.01	0.06	0.01	0.01
*Rubus ulmifolius* Schott.	0.09	0.00	0.00	0.06	0.00	0.00

**Table 3 plants-14-03709-t003:** ^14^C radiocarbon dating of the rose prickles from Almunia de Monteagudo site.

Sample Code	Lab. Code	BP Date, Uncertainty	Cal Years BC/AD Date * (1σ Probability Ranges) (Data in % Peak Area)	Cal Years BC/AD Date * (2σ Probability Ranges) (Data in % Peak Area)	Median 2σ Probability (Data in % Peak Area)
UE 22003	RICH-36708 (015/2019)	186 ± 26 BP	1660 AD–1690 AD (12.9%); 1730 AD–1810 AD (37.9%); 1920 AD–1950 AD (17.4%)	1650 AD–1700 AD (19.6%); 1720 AD–1820 AD (51.1%); 1830 AD–1880 AD (1.0%); 1910 AD–1955 AD (23.8%)	1720–1820 AD (51.1%)
UE 91003	RICH-36709 (003/2023)	91 ± 26 BP	1690 AD–1730 AD (21.8%); 1810 AD–1840 AD (19.8%); 1870 AD–1920 AD (26.5%)	1690 AD–1730 AD (26.0%); 1800 AD–1930 AD (69.4%)	1800–1930 AD (69.4%)

* M, Stuiver, PJ, Reimer, R, Reimer: ^14^C Calibration Program, v.8.2, 2025 [57,58].

## Data Availability

The original contributions presented in the study are included in the article and Appendix A; further inquiries can be directed to the corresponding authors.

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
