# Peer review of "Advancing Archaeobotanical Methods: Morphometry, Bayesian Analysis and AMS Dating of Rose Prickles from Monteagudo Almunia, Spain (12th Century–Present)"

_plants, 2025, doi:10.3390/plants14243709_

Round 1
Reviewer 1 Report
Comments and Suggestions for Authors
Dear authors, I greatly appreciated the originality of your manuscript and the scientific presentation of the results of this study. I have nothing to comment on. The manuscript can be published in this form.
Author Response
- Dear authors, I greatly appreciated the originality of your manuscript and the scientific presentation of the results of this study. I have nothing to comment on. The manuscript can be published in this form.
- Dear reviewer we highly appreciate your positive evaluation of our manuscript.
- We revised figures and tables in order to improve readability.
Reviewer 2 Report
Comments and Suggestions for Authors
The paper is original, accurate, comprehensive, and necessary. Indeed, studies on spines (prickles or thorns) are extremely rare in Archaeobotany, so it's important that this work be published. Congratulations to the authors!
Just two editorial suggestions:
- I would remove "(Accelerator Mass Spectrometry)" from the title.
- I would reduce/simplify the keywords.
Author Response
Reviewer 2:
The paper is original, accurate, comprehensive, and necessary. Indeed, studies on spines (prickles or thorns) are extremely rare in Archaeobotany, so it's important that this work be published. Congratulations to the authors!
- Many thanks for reading and reviewing our manuscript!
Just two editorial suggestions:
1
- I would remove "(Accelerator Mass Spectrometry)" from the title.
- Thanks! Done, we included a row in the abbreviations section with the equivalence of AMS
2
- I would reduce/simplify the keywords.
- Many thanks! Done.
Reviewer 3 Report
Comments and Suggestions for Authors
Dear Authors,
Cultural development of the Iberian Peninsula was highly complex, and these unusual patterns should be reflected by the local plant history. This has remained an urgent topic for the contemporary international research. Your work deals with the rose cultivation in the 12th century and concludes about the continuity and the longevity of the related practices, as well as it emphasizes on the role of al-Andalus in the horticulture developments. The topic of the manuscript is very interesting and, undoubtedly, internationally important. The objective is clear. You explain the methodology in detail, and, thus, the readers can understand what and how was done. Importantly, the limitations are noted. The results are reasonable, and the interpretations are highly interesting and multiple. The conclusions are formulated clearly, and they follow directly from what is written in the main body of the manuscript. The illustrations are rich, accurate, and suitable. I especially like the historical images. The manuscript is referenced adequately. I see that the storytelling is logical, and the writing is easy to follow. Linguistically, the text is ok. Generally, this is a very strong and perfect work, and this can become a good example for how the outcomes of such studies should be reported.
Author Response
Cultural development of the Iberian Peninsula was highly complex, and these unusual patterns should be reflected by the local plant history. This has remained an urgent topic for the contemporary international research. Your work deals with the rose cultivation in the 12th century and concludes about the continuity and the longevity of the related practices, as well as it emphasizes on the role of al-Andalus in the horticulture developments. The topic of the manuscript is very interesting and, undoubtedly, internationally important. The objective is clear. You explain the methodology in detail, and, thus, the readers can understand what and how was done. Importantly, the limitations are noted. The results are reasonable, and the interpretations are highly interesting and multiple. The conclusions are formulated clearly, and they follow directly from what is written in the main body of the manuscript. The illustrations are rich, accurate, and suitable. I especially like the historical images. The manuscript is referenced adequately. I see that the storytelling is logical, and the writing is easy to follow. Linguistically, the text is ok. Generally, this is a very strong and perfect work, and this can become a good example for how the outcomes of such studies should be reported.
- Dear reviewer we greatly appreciate your analysis of our papaer and are pleased you did find it as interesting and well organized